# A Comparative Study on Skid Resistance of Concrete Pavements Differing in Texturing Technique

**DOI:** 10.3390/ma14010178

**Published:** 2021-01-01

**Authors:** Pawel Gierasimiuk, Marta Wasilewska, Wladyslaw Gardziejczyk

**Affiliations:** Faculty of Civil Engineering and Environmental Science, Bialystok University of Technology, 15-351 Bialystok, Poland; marta.wasilewska@pb.edu.pl (M.W.); w.gardziejczyk@pb.edu.pl (W.G.)

**Keywords:** skid resistance, concrete pavement, British pendulum tester, mictotexture, macrotexture

## Abstract

The paper presents a comparison of the skid resistance of concrete pavements textured with different techniques in the process of simulating phenomena occurring in actual road conditions. Tests were carried out on five different texturing methods for concrete pavements: burlap drag (BuD), brush drag (BrD), transverse tining (TT), longitudinal tining (LT) and exposed aggregate concrete (EAC). Changes in the skid resistance were recorded by measurements with a British pendulum tester and a circular texture meter before and during the simulation of the abrasion (1st phase of test) and polishing (2nd phase of test) of specimens using a slab polisher. The results of BPN (British pendulum number) and MPD (mean profile depth) allowed us to determine the influence of microtexture and macrotexture on skid resistance. Analysis of variance showed that the method of texturing concrete pavements has a significant influence on the mean BPN values as well as the MPD parameter at each stage of the test. In order to distinguish homogeneous groups in terms of BPN and MPD levels at the particular stages of the process, the Tukey’s HSD (honest significant difference) post-hoc test was performed. It was found that EAC obtained the most favorable results of all the tested pavement types. Due to the high value of the MPD coefficient after the test and the appropriate values of the friction coefficient, it was confirmed that the EAC pavement will be a durable solution due to the guarantee of skid resistance on high-speed roads during its service life.

## 1. Introduction

Skid resistance is one of the features of the road surface that significantly affects the safety of users. It is defined as the characterization of the friction of a road surface when measured in accordance with a standardized method. Numerous studies have shown that the micro- and macrotexture of the pavement significantly affects the skid resistance. In the case of concrete pavements, it is related to the surface texturing technique and the characteristics of the used concrete mix. Pavements textured with burlap drag and brushes are characterized by a low resistance to the wear caused by vehicle wheels during the period of use [1,2]. Transverse and longitudinal tines of pavements provide deeper irregularities and higher durability than those obtained by jute fabric or brush texturing. The more advantageous techniques for texturing concrete pavements, both damaged and newly constructed, include the exposed aggregate concrete (EAC) method, the grinding and grooving method and the next generation concrete surface (NGCS) [3,4,5,6,7].

The EAC method is based on delayed cement hydration and the removal of unbound cement mortar with a mechanical brush or water under pressure. This exposes the coarse aggregate particles that protrude slightly from the top surface [8,9,10]. Concrete pavements with exposed aggregate are made in two layers, in the “wet on wet” system [11]. The top layer should be made of concrete mix with a maximum grain size of 8 or 11 mm [8,9]. In the case of this type of texturing, it is important to use an aggregate with the appropriate resistance to polishing [12,13]. In order to verify the correctness of the macrotexture, the criteria for the number of exposed coarse aggregate grains on an area of 25 cm^2^ were established; the works were conducted mainly in Germany and Austria [8,9,12,14,15]. However, this method of evaluation is time-consuming. The measurement of the parameters describing the macrotexture is easier to perform in order to control the quality of the pavement surface.

The texturing technique using diamond blades (diamond grinding), in addition to leveling longitudinal irregularities, improves the micro- and macrotexture by producing small irregularities and grooves. The next generation concrete surface (NGCS) is the best known method which uses diamond blades [16,17,18,19,20]. This method allows one to obtain a negative texture, and thanks to the optimal placement of the blades, it provides a durable macrotexture and a high level of microtexture [21]. The macrotexture and microtexture of the surface determine the skid resistance of concrete pavements. Maintaining the required level of skid resistance allows one to reduce the accident rate on wet surfaces even up to 70% [22,23]. This is confirmed by studies presenting analyses describing the relationship between the number of road incidents and the decrease in the friction coefficient in adverse weather conditions [22,24,25,26,27].

Macrotexture depends on the shape, size and gradation graining of the aggregate, and has an effect on skid resistance, especially at high vehicle speed. The MPD (mean profile depth) value is the most frequently used parameter describing the surface macrotexture. The use of laser profilographs, which enable measurements in real traffic conditions, allows for a quick assessment of the macrotexture of the upper layers during the pavement’s use. A high macrotexture allows water to drain away through a system of external voids and reduces the aquaplaning phenomenon. On the other hand, microtexture is responsible for breaking the water film at the contact zone between the tire and the road surface, where a dry contact zone is created [28,29,30]. It is important at all speeds. Microtexture depends on the polishing resistance of the coarse aggregate and the content of fine aggregate in the asphalt mixture [31]. The parameter indirectly describing the microtexture is the value of the friction coefficient determined with the use of dynamic devices from the locked-wheel tester group, or stationary devices such as the British pendulum tester or DFT [32,33].

Studies on the skid resistance of concrete pavements were mainly based on the results obtained on in situ road sections [34,35,36,37,38,39]. The variety of dynamic devices for measuring the friction coefficient used in different countries is an issue. Works on harmonizing these measurement methods in Europe are well advanced now. The effects of the last Common Scale elaborations under the ROSANNE project were verified during two European Pavement Friction Workshops (EPFW) held in Nantes, France, in 2017 and 2019. Additionally, an effort has been made to carry out studies to develop the harmonization of dynamic devices based on the IFI (international friction index) using the reference CTM and DFT stationary devices [40]. The lack of a common scale for all the dynamic devices that are used in the world significantly hinders the exchange of information related to the skid resistance of road surfaces.

It should be noted that not all the devices used in the world to assess skid resistance were considered in the ROSANNE and EPFW projects. Therefore, in order to compare different types of pavements, devices for the laboratory assessment of skid resistance are very helpful, because they guarantee the same conditions representing the factors influencing the changes in macrotexture and microtexture in real road conditions. Thanks to them, it is possible to create a ranking of the materials intended for the upper layers of pavements in terms of their skid resistance. Many research approaches have been created that enable the simulation of the phenomena responsible for these changes. A well-known device is the German FAP machine (also called Wehner/Schuzle machine). However, works on developing a relationship between the results obtained in the laboratory and for in-service pavements are still ongoing [13,36,37,41,42]. Numerous polishing stations have also been created around the world, where tires roll on samples prepared from materials intended for the wearing course. The number of wheel passes, the load applied, or the addition of water or abrasive is optional depending on the test procedure developed. Skid resistance is very often assessed by measurements using stationary devices, such as the British pendulum tester or DFT. Due to the fact that the results of the measurements with these devices are sensitive to microtexture, the parameter describing the macrotexture is measured in order to comprehensively assess the skid resistance of the tested surfaces [43,44,45,46]. The assessment of the skid resistance, detailing the changes related to the microtexture and the macrotexture of the tested surfaces, is very valuable information at the planning and design stage of the upper layers of road pavements.

The aim of this study is to compare the skid resistances of cement concrete pavements textured with various techniques in a process simulating the phenomena occurring in the real conditions on roads.

## 2. Materials and Methods

### 2.1. Preparation of Concrete Pavement Specimens

The tests were performed on concrete slabs textured with the following methods:burlap drag (longitudinally to the road axis) (BuD);brush drag (in the direction perpendicular to the road axis) (BrD);transverse tined (in the direction perpendicular to the road axis) (TT);longitudinal tined (in the direction parallel to the road axis) (LT);exposed aggregate concrete pavement (EAC).

In the case of the EAC, two types of aggregate, differing in maximum grain size, were used: 0/8 mm (EAC8) and 0/11 mm (EAC11). The remaining concrete pavement samples, textured with the BuD, BrD, TT and LT methods, were made with aggregate 0/16 mm. A trachybasalt was used as coarse aggregate.

Figure 1 shows the particle size distribution curves of the aggregate used to make concrete slabs for particular texturing techniques.

Table 1 and Table 2 provide the characteristics of the designed concrete mixtures, and the properties of the concrete mixtures and hardened concrete.

The dimensions of the test slabs were 35 cm × 35 cm × 5 cm. They were compacted on a vibrating table. Six slabs were made for each texturing technique, of which four were randomly selected for testing. Each of the slabs was prepared in a separate technological process.

Transverse and longitudinal tining, and burlap and brush drag were performed on the fresh concrete mixtures, immediately after compacting each slab. The EAC method was carried out in two stages—immediately after compaction, the top surface of the slab was sprayed with a hydration-retarding agent based on paraffin wax and polyalcohol, and after about 24 h the non-hydrated layer of cement mortar was removed with a steel brush, which left the exposed grains of coarse aggregate. Figure 2 shows images of the top surfaces of the tested concrete slabs.

### 2.2. Test Methods

Concrete slabs were subjected to abrasion and polishing in laboratory conditions. This was a simulation of the phenomena occurring in real conditions on the surfaces of road pavements under the influence of pollutants, water and car traffic. For this purpose, the slab polisher was used (Figure 3). The slab polisher is a proprietary device built at the Bialystok University of Technology (Bialystok, Poland). The machine consists of three smooth rubber wheels mounted on a rotating disk and rolling on the surfaces of the specimens. The polishing wheels move at the velocity of 50 ± 2 rotations per minute. The test lasted 6 h and consisted of two three-hour phases. In the first phase water and coarse emery (300/600 µm) were fed continuously onto the surfaces and in the second phase water and emery flour (<53 µm) were fed. A more detailed description of the slab polisher and examples of the results of the research can be found in the literature [9,54].

Skid resistance was determined on the basis of BPN measurements in accordance with ASTM E303-93 [55] (Figure 4a). The shift length of the rubber slider was 126 ± 2mm (Figure 4b). The measurements were made in the traces left by the polishing wheels on the surfaces of the specimens before and after the individual phases. Three replicate passes were performed on each tested specimen. Before each measurement using the British pendulum tester, each specimen was thoroughly washed—corundum and fine pollutions were removed. This procedure is also performed in other devices of this type, e.g., in the FAP device (known as Wehner/Schulze, Wennigsen, Germany).

The assessment of the macrotexture of the concrete slabs was performed by measuring the MPD parameter with the circular texture meter in accordance with ASTM E2157-15 (2019) [56] (Figure 5a). Due to the fact that the CTM scans the profile along a perimeter divided into 8 segments, in the cases of slabs with directional texturing (Bud, BrD, TT, LT), the MPD values were read only from the appropriate parts of the tested surfaces (Figure 5b).

On each of the test slabs, BPN measurements were taken at three points prior to the polishing and during the process:in phase I after 5, 10, 15, 30, 60, 90, 120, 150 and 180 min;in phase II after 185, 190, 195, 210, 240, 270, 300, 330 and 360 min.

Measurements of the MPD parameter were takenon each of the concrete slabs before the process was started, after phase I and after phase II, over three repetitions.

## 3. Results and Discussion

### 3.1. Changes in Skid Resistance during the Polishing Process

#### 3.1.1. Changes in BPN Value

Table 3 shows the descriptive statistics of BPN values at different stages of polishing (mean value ($\bar{\mathrm{BPN}}$), standard deviation (STD), coefficient of variation (V), minimum and maximum (BPN_min_, BPN_max_)).

Specimens with the transverse texturing direction, BrD and TT, were characterized by high BPN values compared to other tested technologies (Figure 6b,c). This is due to the fact that the direction of the pendulum arm was perpendicular to the groove system. On the other hand, in the case of the LT surface (Figure 6d), the direction of the pendulum arm during the measurement was parallel to the groove system. The circumstances of the measurement in the laboratory were also due to the texture pattern of the actual road conditions. Consequently, the BrD and TT texturing techniques allow one to achieve higher BPN values compared to LT. The BuD surface reached low BPN values at the end of the test, comparable to the LT values.

EAC pavements do not have a directed texture (Figure 6e,f). Higher BPN values in the polishing process were achieved by the EAC11 surface, which reached higher values at each stage of polishing.

After the first phase of the process, in the cases of pavements with the texture types BuD, BrD, TT and LT, a decrease of 5–13% in the BPN values was recorded compared to the initial values. In the cases of BrD and TT slabs, the BPN values changed at a similar level (Figure 7). This was influenced by the lateral orientation of the grooves (TT) and the unevenness resulting from the lateral brushing (BrD) in relation to the direction of operation of the polishing wheels in the slab polisher.

The BPN values for the EAC samples decreased by 21% and 18% after the first phase, and by 12% and 9% in the second phase (EAC8 and EAC11, respectively). In the case of the rest of the tested specimens, decreases in the BPN values ranging from 12 to 25% were recorded in phase II.

Based on the values recorded during the particular phases, interesting tendencies were noticed when subjecting the samples to factors related to the simulation of phenomena occurring in the real conditions. Changes in the BPN values, along with standard deviations recorded in the particular test phases, are shown in Figure 8.

In real traffic conditions, surfaces are exposed to polishing, abrasion and weathering as a result of car traffic, water, pollution and weather conditions [57]. Due to the fact that these phenomena occur simultaneously but with varying intensity depending on climatic conditions, it is difficult to reproduce them in laboratory conditions. In the case of the slab polisher, it was determined that the addition of coarse emery (300/600 µm) would contribute to conditions favorable to abrasion, while emery flour (<53 µm) to polishing. Such test conditions are those under which the polishing resistance of coarse aggregates is assessed according to EN 1097-8. The variation in BPN values varied depending on the texturing method. For BrD, TT, LT and EAC, a slight decrease in BPN was observed during the first 30 min. After 60 min, this value stabilized at a certain level and only insignificant differences were recorded until the end of phase I. This is related to changes in the microtextures of surfaces. Coarse emery, which is spread on the surface under the movement of wheels, rubs the cement mortar, removes it from the surface and creates micro-grooves. As a consequence, the minerals changed their original appearance. The tests conducted on coarse aggregates have shown that in this phase, polished surfaces with a characteristic gloss are also observed; however, the surfaces of numerous abrasion marks predominate [58]. Slight differences in BPN values showing both decreasing and increasing trends appeared in phase I after 60 min, which were related to the effects of the polishing and abrasion of the surface. While the trends of these changes were visible on the BrD, TT, LT and EAC surfaces, the decreases and increases in BPN were not large. In the cases of the BuD surfaces, the recorded decreases and increases in BNP were significant. After 180 min, the BPN value was comparable to the 30 min test result.

On all tested surfaces, due to the presence of emery flour and water, lower final values were obtained compared to the values after the end of phase I. This decrease is related to the smoothing of the surface due to its polishing.

In order to recognize the mechanisms responsible for BPN changes during polishing, images of the surfaces before and after the test were analyzed under an optical-digital microscope. In the case of jute dragging-textured slabs, the polishing process removed the top layer of cement mortar. At this stage, the cement mortar with exposed fine aggregate is mainly responsible for the skid resistance (Figure 9a). In the case of tined and brushed slabs, the grooves and unevenness were damaged during polishing, which resulted in a decrease in the BPN values (Figure 9b–d). The values of the friction coefficient were mainly influenced by the unevenness resulting from the texturing method and the presence of fine aggregate exposed in the cement mortar. Fine aggregate grains were also visible on the surfaces.

Coarse aggregate plays a more important role in EAC surfaces. Cement mortar was removed from the surfaces of the aggregate grains as a result of polishing (Figure 9e,f), and the exposed coarse aggregate grains could be treated with polishing agents. It should be noted that the contact surface on the aggregate often amounted to less than 50% of the surface of the protruding grain [59].

#### 3.1.2. Changes in Surface Macrotexture

Table 4 presents descriptive statistics calculated on the basis of the MPD measurement results—mean value ($\bar{\mathrm{MPD}}$), standard deviation (STD), coefficient of variation (V), and minimum and maximum (MPD_min_ and MPD_max_) values. Figure 10 shows the profiles of surface roughness, before polishing and after phase II.

The lowest MPD values were recorded on the slabs textured by burlap drag, with an MPD at the level of about 0.25 mm. No changes were noticed in the texture profiles between the initial state and after polishing (Figure 10a).

The greatest differences in MPD values during polishing (decrease by approx. 65%) were recorded on the surfaces textured by a brush drag (BrD). This was due to the low resistance of pavements textured by this method to factors related to movement. Irregularities created by the brush were removed from the surface (Figure 10b). Significant changes in the texture profiles were recorded on the surfaces grooved transversely and longitudinally—TT and LT (Figure 10c,d).

The highest MPD values were obtained on the EAC slabs. These surfaces had similar profiles both before and after the test (Figure 10e,f). The differences between before the test and after the completion of phase II were caused by the removal of the cement mortar and the polishing of the coarse aggregate. The lowest values of the MPD coefficient of variation, ranging from 11% to 22%, were obtained for the EAC8 and EAC11 specimens. The other texturing methods had coefficients of variation ranging from 17% to 36%. This indicates a high variability of MPD with respect to individual textures (Table 4).

The EAC surfaces were the most resistant to the processes taking place during the polishing test. Despite the decrease in the MPD value during the polishing process, the final values were over 0.8 mm (Table 4). Low MPD values after the polishing process were recorded on the burlap drag, the brush drag, and transverse and longitudinal tined surfaces: BuD = 0.26 mm, BrD = 0.25 mm, TT = 0.47 mm, LT = 0.30 mm.

### 3.2. The Influence of the Texturing Method on the BPN and MPD Coefficients in the Polishing Process

One-way analysis of variance (Factor A—texturing technique) was used at the particular polishing stages to determine the influence of the texturing methods of concrete pavements on their skid resistances, described by the BPN and MPD parameters. The STATISTICA 13.1 program was used for the calculations.

Table 5, Table 6, Table 7, Table 8, Table 9 and Table 10 show the results of the variance analyses of the BPN and MPD values obtained at the particular stages of the test. Important parameters are marked in red color.

At the significance level α = 0.05, a significant influence of factor A (the method of texturing concrete pavements) on the mean BPN values and the MPD parameter was found both before starting the tests on the slab polisher and after each phase of the tests.

The graphic interpretation of the obtained results of the analysis of variance for BPN and MPD values at the particular stages of polishing, in relation to the analyzed methods of texturing concrete pavements, is presented in Figure 11 and Figure 12.

There were significant changes affecting the microtextures of the tested surfaces both in phase I and phase II. The intensity of these changes varied at particular stages of the tests, depending on the type of texturing technique. It was found that by far the highest BPN standard deviation values were obtained before the polishing phase in each of the texturing methods (Figure 11). The reason for this phenomenon is the heterogeneity of the microtexture immediately before the test. In the case of the BuD, BrD, LT and TT methods, the fine aggregate grains were exposed on the tested surface, which caused the increasing of the BPN. However, there were places where the texture was smooth (cement mortar), and was characterized by low values of the friction coefficient, hence the high values of standard deviation.

The EAC surfaces immediately after texturing were characterized by the cement mortar remaining after the exposure of the aggregate, which was removed from the aggregates along with the progressing process of polishing.

On the other hand, the macrotextures of these surfaces significantly changed only after the first phase of test, when the conditions simulated the phenomenon of abrasion. The differences between the MPD values of most of the tested surfaces are inconsiderable.

Due to the equal variances between the homogeneous groups in terms of the BPN and MPD levels at the particular stages of the process, an analysis was performed using the Tukey’s HSD post-hoc test. The calculations were performed in STATISTICA 13.1.

Table 11 and Table 12 present the results of the Tukey’s HSD test for the BPN friction coefficient and MPD parameter. The values of the probability level *p* < 0.05 are marked in red. They indicate statistically significant differences between the values of the parameters obtained for the individual texturing methods. Figure 13 shows a graphical interpretation of the results of the Tukey’s HSD test in relation to the individual test phases in the slab polisher.

The slabs with textured perpendicularly to the driving axis (BrD, TT) and EAC did not show any significant differences in terms of the initial BPN values (Table 11). No significant differences in BPN values were recorded after phase I and phase II between the slabs with the same polishing direction (BuD and LT, and TT and BrD). In the case of the EAC surface, only the results after the second phase of the test differ significantly.

Significant differences between MPD values occurred in most of the texturing techniques at each stage of the performed test. The only exceptions are the EAC 8 and EAC 11 pavements. Even though they differed in aggregate size, their roughness profiles were very similar to each other. It was noticed that the BuD, LT and BrD surfaces also showed no significant differences in their mean profile depth values after phases I and II. This is due to the fact that their macrotexture is sensitive when there is a predominance of coarse dirt particles on the surface, which, under the influence of car traffic, rub and destroy the texture made by these methods, while the very small impurities contributing to polishing do not have a destructive effect on the profile depth.

The analysis of changes in the BPN and MPD values of the surfaces textured with various techniques at particular stages of the tests proved that EAC8 and EAC11 are very good technologies for the construction of the wearing course of concrete pavements intended for high-speed roads (above 90 kmph). They are characterized by the highest MPD values compared to other tested surfaces. Due to the exposed coarse aggregate grains, their resistance to polishing agents will be related to the polishing resistance of the rock [60,61]. Therefore, in some countries, this technology has specific requirements for the PSV (Polished Stone Value) of coarse aggregates. The EAC pavements have slightly lower BPN results compared to BrD and TT, but higher results than BuD and LT. Because the profile depths were low after the completion of the test, the BrD, BuD, LT and TT wearing courses are only recommended for lower-speed roads of local importance.

## 4. Conclusions

The application of a laboratory device that simulates the phenomena occurring on the road (abrasion, polishing) allowed us to determine the significant differences between the BPN and MPD parameters for concrete pavements textured with different techniques. Based on the obtained results, the following conclusions were formulated:The least favorable results in relation to BPN and the MPD macrotexture parameter were obtained for the BuD and LT surfaces. This pavement was characterized by the lowest BPN values (51 and 54, respectively) and a poorly developed macrotexture (0.26 mm and 0.30 mm, respectively);The BrD and TT surfaces were characterized by the highest BPN values in the initial period and after finishing the polishing process. However, they obtained very low macrostructure values (0.25 mm and 0.45 mm);The EAC surfaces proved to be the most resistant to conditions simulating the phenomena of abrasion and polishing;Analysis of variance showed significant differences between the BPN and MPD parameters depending on the texturing methods of concrete pavements. The type of texturing has a significant impact on the skid resistance of concrete pavements;The effect of the texturing direction on the skid resistance is shown. In many cases, from the pavements with the same texturing direction, homogeneous groups were created—longitudinal (BuD, LT), transverse (BrD, TT) and non-directional (EAC8, EAC11) (using Tukey’s HSD post-hoc test).

In addition, the experiment allowed us to select the texturing method that will be a permanent solution to guarantee good skid resistance on high-speed roads during their period of use. Among the analyzed techniques for texturing concrete pavements, EAC is the most advantageous solution. This is confirmed by the slight changes in the coefficient of friction in the polishing process, and the final BPN values. The MPD values show that the adoption of the EAC method as the primary technique for texturing concrete pavements would be an appropriate solution for high-speed roads with heavy traffic.

It should be emphasized that the evaluation of skid resistance properties at the stage of selecting a technology for road construction provides valuable information about the potential changes taking place on the wearing course of the road pavement. The analysis of the test results allows for the selection of an appropriate solution, taking into account the conditions related to the site location (junction, straight and curve segments) of the road section.

## Figures and Tables

**Figure 1 materials-14-00178-f001:**
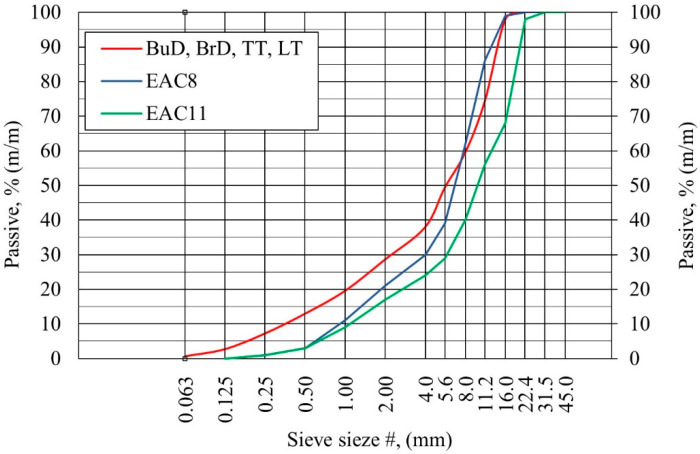
Particle size distribution curves of concrete mixtures.

**Figure 2 materials-14-00178-f002:**
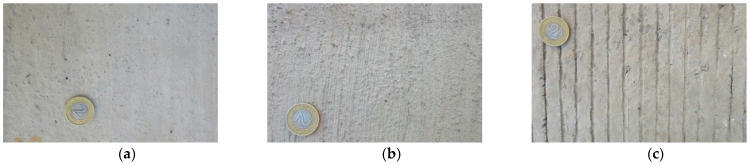
Slab textures: (**a**) BuD, (**b**) BrD, (**c**) TT, (**d**) LT, (**e**) EAC8, (**f**) EAC11 (diameter of the presented coin is 21.5 mm).

**Figure 3 materials-14-00178-f003:**
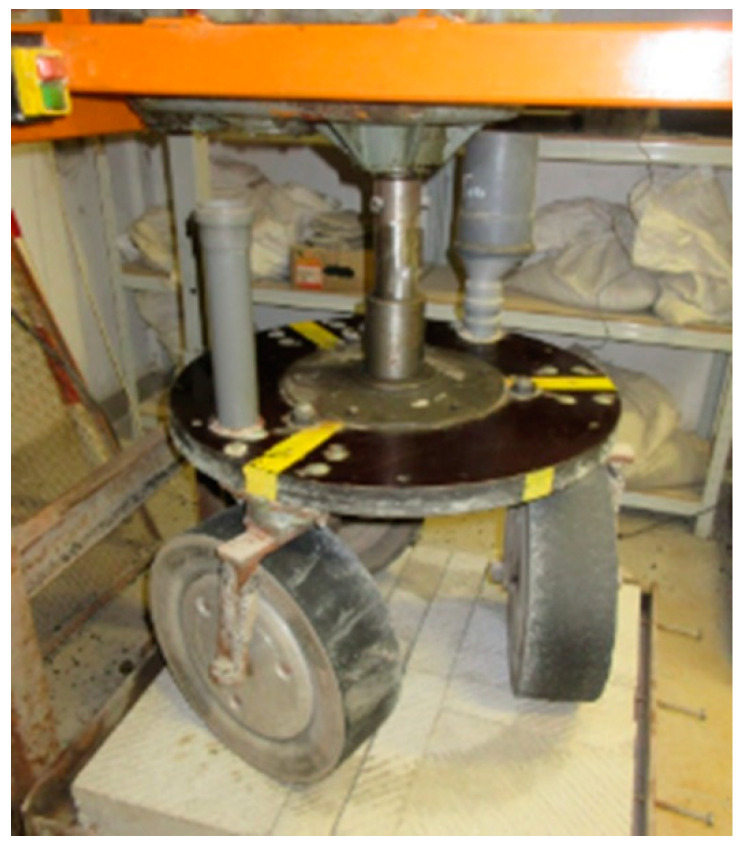
Slab polisher.

**Figure 4 materials-14-00178-f004:**
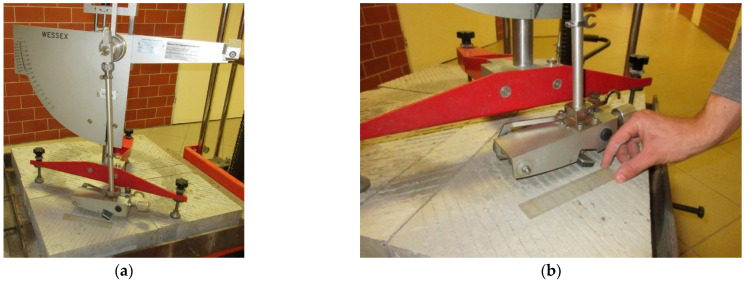
(**a**) Measurement with British pendulum tester. (**b**) Setting the shift length of the rubber slider.

**Figure 5 materials-14-00178-f005:**
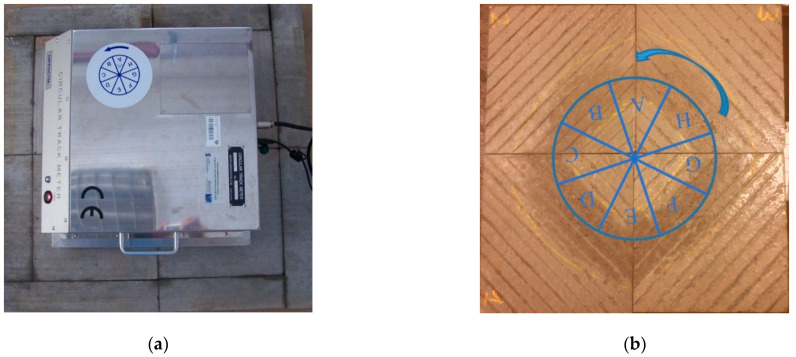
(**a**) Measurement of macrotexture with CTM, (**b**) the arrangement of the segments during the measurement on an example slab LT.

**Figure 6 materials-14-00178-f006:**
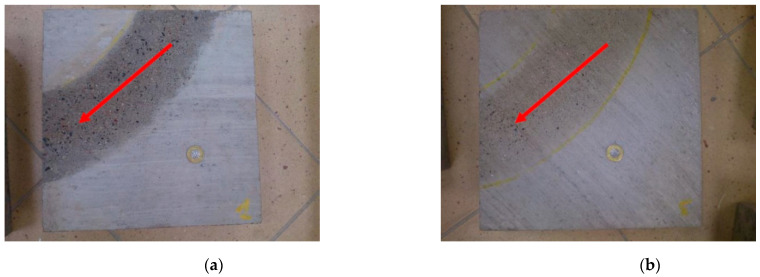
Textures of concrete pavements with the marked measurement direction of the British pendulum tester: (**a**) BuD, (**b**) BrD, (**c**) TT, (**d**) LT, (**e**) EAC8, (**f**) EAC11 (diameter of the presented coin is 21.5 mm).

**Figure 7 materials-14-00178-f007:**
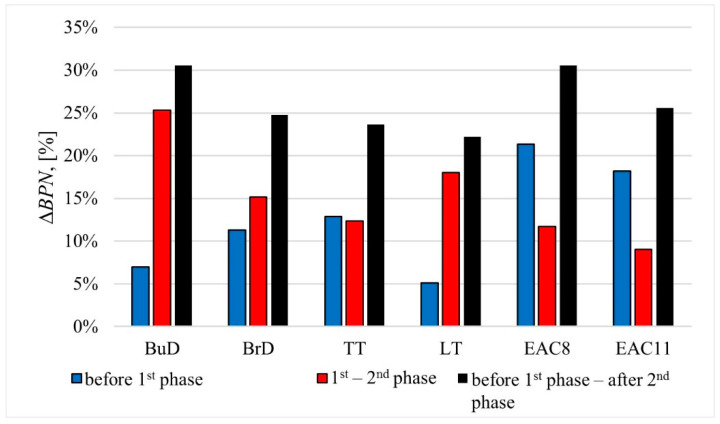
The percentage decrease in BPN between the individual phases of the polishing process.

**Figure 8 materials-14-00178-f008:**
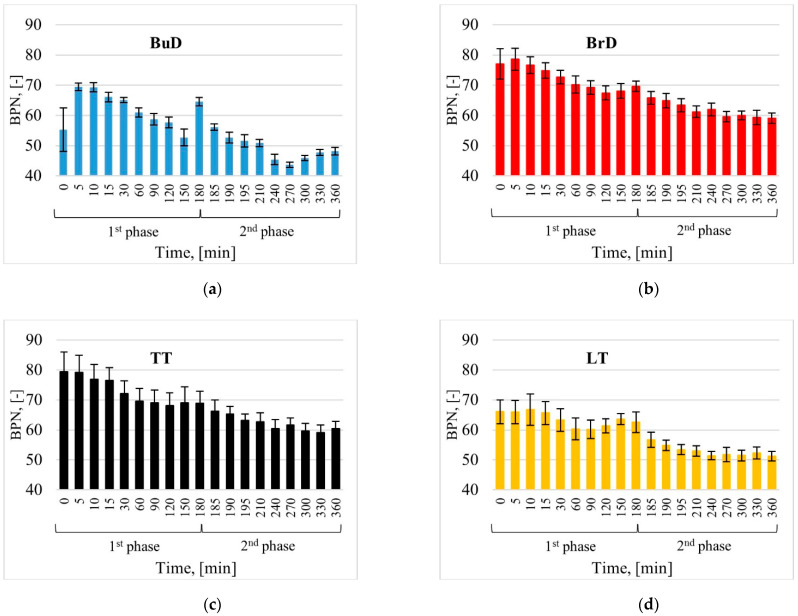
Changes in the mean BPN values of concrete pavements textured by (**a**) burlap drag, (**b**) brush drag, (**c**) transverse tining, (**d**) longitudinal tining, (**e**) EAC8, (**f**) EAC11.

**Figure 9 materials-14-00178-f009:**
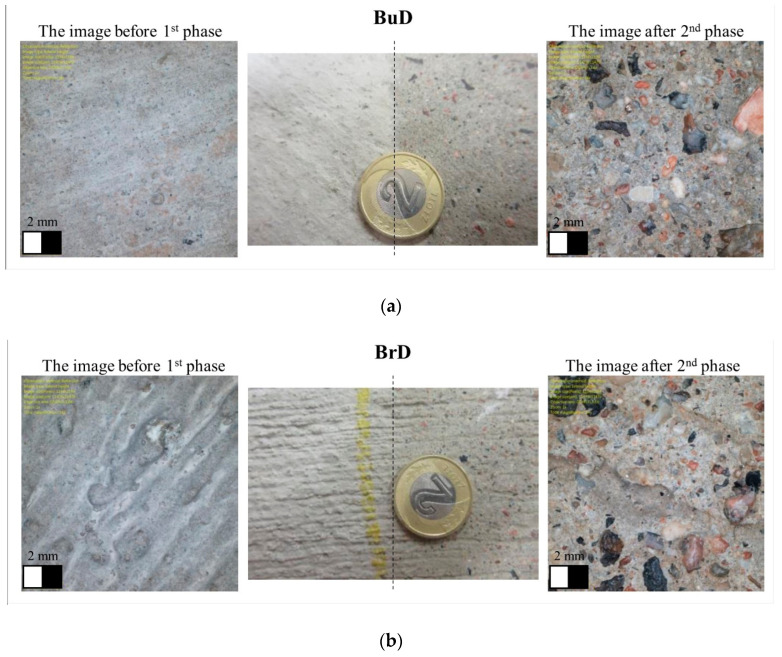
The surfaces of concrete specimens (diameter of the presented coin is 21.5 mm) and images from an optical-digital microscope: (**a**) BuD, (**b**) BrD, (**c**) TT, (**d**) LT, (**e**) EAC8, (**f**) EAC11.

**Figure 10 materials-14-00178-f010:**
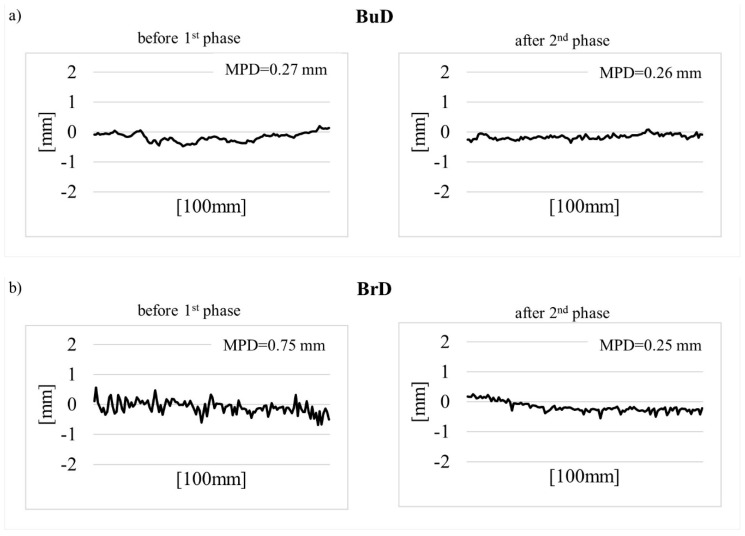
The profiles of the tested surfaces at individual stages of the test: (**a**) BuD, (**b**) BrD, (**c**) TT, (**d**) LT, (**e**) EAC8, (**f**) EAC11.

**Figure 11 materials-14-00178-f011:**
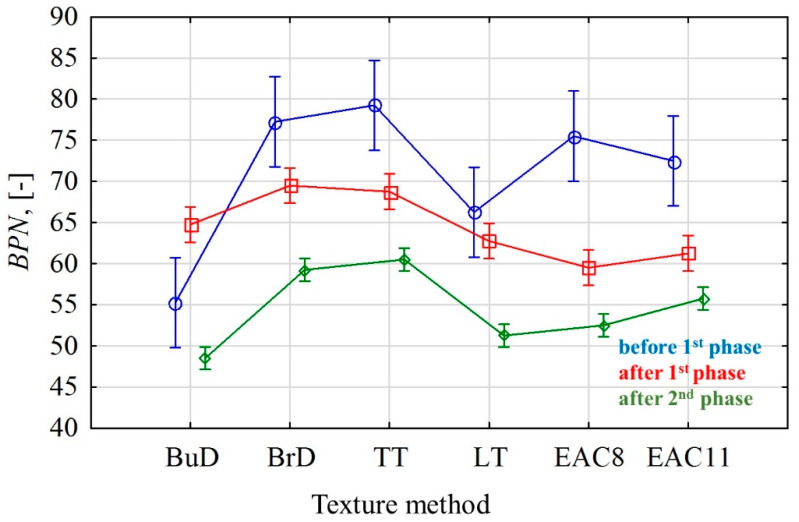
Mean BPN values with the 95% confidence interval.

**Figure 12 materials-14-00178-f012:**
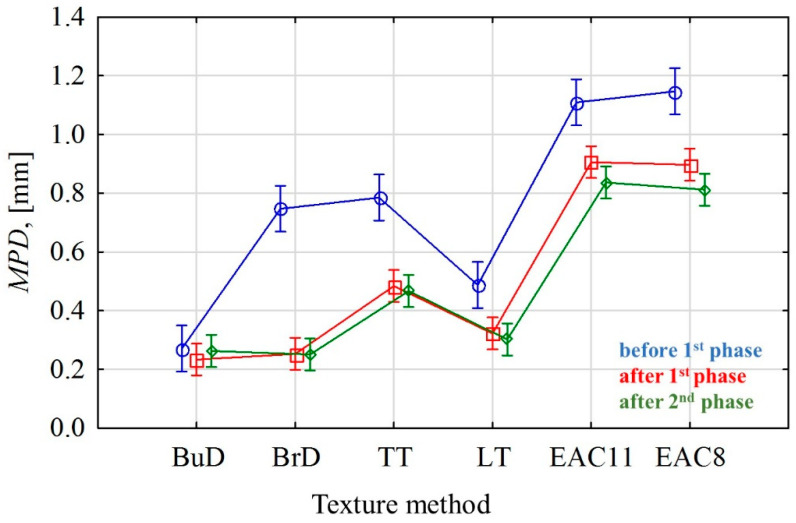
Mean MPD values with the 95% confidence interval.

**Figure 13 materials-14-00178-f013:**
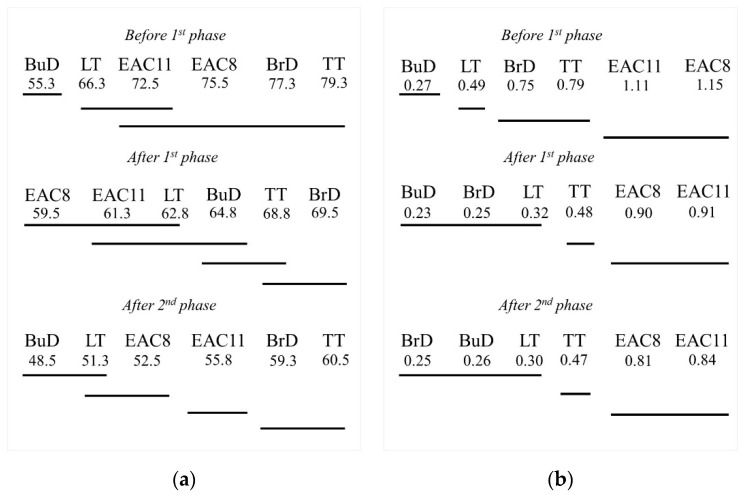
Homogeneous groups according to the HSD Tukey’s test for mean values of (**a**) BPN, (**b**) MPD.

**Table 1 materials-14-00178-t001:** Characteristics of the designed concrete mixtures.

Property	BrD; BuD; TT; LT	EAC8	EAC11
Strength class	C35/45	C35/45	C35/45
Consistency class acc. to PN-EN 12350-3 [47]	V2	V2	V2
Exposure class	XF4	XF4	XF4
D_max_ (mm)	16	8	11
Cement type and class	CEM I 42.5 R	CEM I 42.5 R	CEM I 42.5 R
Cement (kg/m^3^)	400.0	440.0	440.0
w/c	0.36	0.38	0.38
Fine aggregate	0/2	0/2	0/2
Coarse aggregate	2/5, 4/8, 8/11, 11/16	2/5, 4/8	2/5, 4/8, 8/11
Admixture 1	Air-entraining	Air-entraining	Air-entraining
Admixture 2	Water reducing admix. based on polycarboxylates and phosphonates	Water reducing admix. based on polycarboxylates and phosphonates	Water reducing admix. based on polycarboxylates and phosphonates

**Table 2 materials-14-00178-t002:** Properties of concrete mixtures and hardened concrete with trachybasalt coarse aggregate.

Property	BrD; BuD; TT; LT	EAC8	EAC11
Air content acc. to PN-EN 12350-7 [48] (%)	5.0	5.0	5.1
Density acc. to PN-EN 12350-6 [49] (kg/m^3^)	2433	2453	2410
Density acc. to PN-EN 12390-7 [50] (kg/m^3^)	2380	2413	2376
Compressive strength acc. to PN-EN 12390-3 [51] (MPa)	59.0	60.5	50.5
Flexural strength acc. to PN-EN 12390-5 [52] (MPa)	6.9	8.0	7.4
Freezing/thawing in the presence of deicing agents, freeze resistance category acc. to PN-EN 13877-2 [53]	FT2	FT2	FT2

**Table 3 materials-14-00178-t003:** Descriptive statistics of BPN values of concrete pavements.

Texturing Method	Stage of Test	$\bar{BPN},\;(–)$	STD, (–)	V, (%)	BPN_min_, (–)	BPN_max_, (–)
BuD	Before phase I	55	7.2	13%	44	67
	After phase I	65	1.4	2%	61	66
	After phase II	48	1.3	3%	46	51
BrD	Before phase I	77	5.1	7%	70	86
	After phase I	70	1.7	3%	66	73
	After phase II	59	1.7	3%	55	61
TT	Before phase I	79	6.6	8%	70	90
	After phase I	69	3.9	6%	65	81
	After phase II	60	2.5	4%	56	65
LT	Before phase I	66	4.0	6%	60	76
	After phase I	63	3.4	5%	55	68
	After phase II	51	1.6	3%	48	54
EAC8	Before phase I	76	3.6	5%	69	81
	After phase I	59	0.9	1%	58	61
	After phase II	52	1.3	2%	50	55
EAC11	Before phase I	73	2.6	4%	70	80
	After phase I	61	1.2	2%	59	64
	After phase II	56	1.5	3%	53	58

**Table 4 materials-14-00178-t004:** Descriptive statistics of MPD.

Texturing Method	Stage of Test	$\bar{MPD},\;(\mathrm{mm})$	STD, (mm)	V, (%)	MPD_Min_, (mm)	MPD_Max_, (mm)
BuD	Before phase I	0.27	0.1	36%	0.18	0.55
	After phase I	0.23	0.06	26%	0.15	0.40
	After phase II	0.26	0.09	33%	0.13	0.40
BrD	Before phase I	0.75	0.17	23%	0.51	1.06
	After phase I	0.25	0.06	23%	0.17	0.40
	After phase II	0.25	0.08	32%	0.11	0.40
TT	Before phase I	0.80	0.13	17%	0.56	0.97
	After phase I	0.47	0.11	23%	0.34	0.78
	After phase II	0.47	0.12	26%	0.31	0.78
LT	Before phase I	0.49	0.14	29%	0.32	0.74
	After phase I	0.32	0.11	36%	0.17	0.45
	After phase II	0.30	0.11	37%	0.15	0.45
EAC8	Before phase I	1.15	0.26	22%	0.85	1.57
	After phase I	0.90	0.17	19%	0.62	1.11
	After phase II	0.81	0.14	17%	0.61	1.11
EAC11	Before phase I	1.11	0.13	11%	0.92	1.35
	After phase I	0.91	0.12	13%	0.73	1.16
	After phase II	0.84	0.13	16%	0.67	1.16

**Table 5 materials-14-00178-t005:** Results of analysis of variance (BPN)—before test.

Effect	Sum of Squares, SS	Degrees of Freedom, df	Mean Sum of Squares, MS	*F_A,obl_*	*p*-Value
A	1601.00	5	320.20	11.79	0.00
Error	489.00	18	27.20	-	-
Sum	2090.00	23	-	-	-

**Table 6 materials-14-00178-t006:** Results of analysis of variance (MPD)—before test.

Effect	Sum of Squares, SS	Degrees of Freedom, df	Mean Sum of Squares, MS	*F_A,obl_*	*p*-Value
A	9.39	5	1.88	75.22	0.00
Error	2.25	90	27.20		
Sum	11.63	95			

**Table 7 materials-14-00178-t007:** Results of analysis of variance (BPN)—after phase I.

Effect	Sum of Squares, SS	Degrees of Freedom, df	Mean Sum of Squares, MS	*F_A.obl_*	*p*-Value
A	326.03	5	65.37	15.69	0.00
Error	75.00	18	4.17		
Sum	401.83	23			

**Table 8 materials-14-00178-t008:** Results of analysis of variance (MPD)—after phase I.

Effect	Sum of Squares, SS	Degrees of Freedom, df	Mean Sum of Squares, MS	*F_A,obl_*	*p*-Value
A	7.76	5	1.55	129.59	0.00
Error	1.08	90	0.01		
Sum	1.28	95			

**Table 9 materials-14-00178-t009:** Results of analysis of variance (BPN)—after phase II.

Effect	Sum of Squares, SS	Degrees of Freedom, df	Mean Sum of Squares, MS	*F_A,obl_*	*p*-Value
A	442.38	5	88.48	50.96	0.00
Error	31.25	18	1.74		
Sum	473.63	23			

**Table 10 materials-14-00178-t010:** Results of analysis of variance (MPD)—after phase II.

Effect	Sum of Squares, SS	Degrees of Freedom, df	Mean Sum of Squares, MS	*F_A,obl_*	*p*-Value
A	5.90	5	1.18	97.64	0.00
Error	1.09	90	0.01		
Sum	6.98	95			

**Table 11 materials-14-00178-t011:** The results of the Tukey’s HSD test calculations for mean BPN values.

Texturing Method	BuD	BrD	TT	LT	EAC8	EAC11
Before test						
BuD						
BrD	0.000					
TT	0.000	0.994				
LT	0.073	0.073	0.025			
EAC8	0.001	0.997	0.906	0.173		
EAC11	0.002	0.787	0.472	0.551	0.961	
After phase I						
BuD						
BrD	0.040					
TT	0.109	0.995				
LT	0.735	0.002	0.007			
EAC8	0.020	0.000	0.000	0.263		
EAC11	0.200	0.000	0.001	0.898	0.825	
After phase II						
BuD						
BrD	0.000					
TT	0.000	0.759				
LT	0.078	0.002	0.000			
EAC8	0.005	0.000	0.000	0.759		
EAC11	0.000	0.015	0.001	0.002	0.027	

**Table 12 materials-14-00178-t012:** The results of the Tukey’s HSD test calculations for mean MPD values.

Texturing Method	BuD	BrD	TT	LT	EAC8	EAC11
Before test						
BuD						
BrD	0.000					
TT	0.000	0.984				
LT	0.003	0.000	0.000			
EAC8	0.000	0.000	0.000	0.000		
EAC11	0.000	0.000	0.000	0.000	0.985	
After phase I						
BuD						
BrD	0.996					
TT	0.000	0.000				
LT	0.208	0.476	0.001			
EAC8	0.000	0.000	0.000	0.000		
EAC11	0.000	0.000	0.000	0.000	0.999	
After phase II						
BuD						
BrD	0.999					
TT	0.000	0.000				
LT	0.912	0.774	0.001			
EAC8	0.000	0.000	0.000	0.000		
EAC11	0.000	0.000	0.000	0.000	0.989	

## Data Availability

The data presented in this study are available in the database of the authors at the Faculty of Civil Engineering and Environmental Science.

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
