# Peer review of "A Comparative Study on Skid Resistance of Concrete Pavements Differing in Texturing Technique"

_materials, 2021, doi:10.3390/ma14010178_

Round 1
Reviewer 1 Report
The paper studies the characteristics and performances of concrete pavements, in terms of macro/microtexture and skid resistance, as a function of the adopted surface texturing techniques. These last can present different problems or advantages: pavements textured with the burlap drag and brushes generally show low resistance to wear caused by vehicle wheels; transverse and longitudinal tines give deeper irregularities and higher durability than those obtained by jute fabric or brush texturing; the EAC method (Exposed Aggregate Concrete) seems to result as one of the most advantageous solutions.
Thus, the aim of the study is to compare the skid resistance of pavements textured with various techniques in a process simulating the wearing phenomena that occur to real roads.
For these purposes, tests were performed on 35 cm x 35 cm x 5 cm concrete slabs, textured as follows: burlap drag (BuD); brush drag (BrD); transverse tined (TT); longitudinal tined (LT); exposed aggregate concrete pavement (EAC). In the case of the EAC, two types of aggregate differing in the maximum grain size were used. Abrasion and polishing processes were applied in laboratory to test slabs, in order to simulate phenomena occurring in real conditions on the surface of road pavements.
The research is not very original but it is well presented and clearly described; the proposed methodology is proper and scientifically correct.
The literature references are quite adequate but it could be furtherly increased. For example, as a suggestion, other studies that could be considered for general or specific facets of the topic are the following:
- Wu, C. L., & Nagi, M. A. (1995). Optimizing surface texture of concrete pavement (Research and development bulletin / PCA Portland Cement Association, No. RD111T) - (book)
that is a comprehensive report useful to summarize many researches and works related to surface texturing technology;
- Loprencipe, G., & Cantisani, G. (2015). Evaluation methods for improving surface geometry of concrete floors: A case study. Case Studies in Structural Engineering, 4, 14-25. (https://doi.org/10.1016/j.csse.2015.06.002)
that considers the techniques to remove longitudinal and transversal irregularities or level the surfaces (including both road pavements and industrial floors) and to measure the results obtained after processing;
- Chen, Y., Wang, K. J., & Zhou, W. F. (2013). Evaluation of surface textures and skid resistance of pervious concrete pavement. Journal of Central South University, 20(2), 520-527.
(https://doi.org/10.1007/s11771-013-1514-y)
that presents fractal geometry and fractal dimension in order to measure the skid resistance on pervious concrete pavement, overcoming the shortcomings of both macrotexture depth and British portable pendulum number.
Regarding to the test method, Authors do not specify if the procedure used to obtain the accelerated abrasion and polishing in laboratory, by means of the Slab Polisher machine, is based on standards and codes and/or it is recognized in the literature and adopted by other authors.
In order to consider the reliability of results, in terms of BPN obtained values, is it certain that these ones represent the effect of surface polishing and they are not determined by contamination? In other terms, is it possible that the slabs after polishing process present on their surface some rubber sediments that can be easily removed by any surfactant cleanser? The performed analysis by optical-digital microscopeare useful to clarify this point?
Regarding to the statistical analysis, very significant differences in the wideness of the 95% confidence interval are present in the comparison between BPN values obtained respectively after Phase I and Phase II, for all the considered texturing methods. How this result can be interpreted? Why this do not happen if considering the BPN values before and after Phase I?
The conclusions actually consist of a small recapitulation of the study; the usefulness and contribution to knowledge of the proposed study should be more clearly presented.
Author Response
Thank you for your insightful and detailed reviews. Our responses and comments are provided below.
The paper studies the characteristics and performances of concrete pavements, in terms of macro/microtexture and skid resistance, as a function of the adopted surface texturing techniques. These last can present different problems or advantages: pavements textured with the burlap drag and brushes generally show low resistance to wear caused by vehicle wheels; transverse and longitudinal tines give deeper irregularities and higher durability than those obtained by jute fabric or brush texturing; the EAC method (Exposed Aggregate Concrete) seems to result as one of the most advantageous solutions.
Thus, the aim of the study is to compare the skid resistance of pavements textured with various techniques in a process simulating the wearing phenomena that occur to real roads.
For these purposes, tests were performed on 35 cm x 35 cm x 5 cm concrete slabs, textured as follows: burlap drag (BuD); brush drag (BrD); transverse tined (TT); longitudinal tined (LT); exposed aggregate concrete pavement (EAC). In the case of the EAC, two types of aggregate differing in the maximum grain size were used. Abrasion and polishing processes were applied in laboratory to test slabs, in order to simulate phenomena occurring in real conditions on the surface of road pavements.
The research is not very original but it is well presented and clearly described; the proposed methodology is proper and scientifically correct.
The literature references are quite adequate but it could be furtherly increased. For example, as a suggestion, other studies that could be considered for general or specific facets of the topic are the following:
- Wu, C. L., & Nagi, M. A. (1995). Optimizing surface texture of concrete pavement (Research and development bulletin / PCA Portland Cement Association, No. RD111T) - (book)
that is a comprehensive report useful to summarize many researches and works related to surface texturing technology;
- Loprencipe, G., & Cantisani, G. (2015). Evaluation methods for improving surface geometry of concrete floors: A case study. Case Studies in Structural Engineering, 4, 14-25. (https://doi.org/10.1016/j.csse.2015.06.002)
that considers the techniques to remove longitudinal and transversal irregularities or level the surfaces (including both road pavements and industrial floors) and to measure the results obtained after processing;
- Chen, Y., Wang, K. J., & Zhou, W. F. (2013). Evaluation of surface textures and skid resistance of pervious concrete pavement. Journal of Central South University, 20(2), 520-527.
(https://doi.org/10.1007/s11771-013-1514-y)
that presents fractal geometry and fractal dimension in order to measure the skid resistance on pervious concrete pavement, overcoming the shortcomings of both macrotexture depth and British portable pendulum number.
Answer: Thank you for your attention. We analyzed the listed papers for inclusion in the paper. We believe that item 3 (Chen, Y., Wang, K. J., & Zhou, W. F. (2013). Evaluation of surface textures and skid resistance of pervious concrete pavement. Journal of Central South University, 20(2), 520-527) is close to our subject and we have included it in the list of references. Item 2 (Loprencipe, G., & Cantisani, G. (2015). Evaluation methods for improving surface geometry of concrete floors: A case study. Case Studies in Structural Engineering, 4, 14-25. (https://doi.org/10.1016/j.csse.2015.06.002)) concerns concrete floors more, while item 1 (Wu, C. L., & Nagi, M. A. (1995). Optimizing surface texture of concrete pavement (Research and development bulletin / PCA Portland Cement Association, No. RD111T) - (book)) is not available to us due to the year of publication (1995). However, we did refer to [1, 2, 4], which also contain interesting information about texturing methods.
Regarding to the test method, Authors do not specify if the procedure used to obtain the accelerated abrasion and polishing in laboratory, by means of the Slab Polisher machine, is based on standards and codes and/or it is recognized in the literature and adopted by other authors.
Answer: The Slab polisher is a proprietary device built at the Bialystok University of Technology. The idea for the device was based on numerous studies carried out in other scientific Institutions. This type of device was created in, among others The United States, Ireland, Australia, and New Zealand. The results of our studies obtained by this method have been repeatedly published (DOI: 10.2478 / v.10169-012-0028-6, DOI: 10.1016 / j.conbuildmat.2020.119921)
In order to consider the reliability of results, in terms of BPN obtained values, is it certain that these ones represent the effect of surface polishing and they are not determined by contamination? In other terms, is it possible that the slabs after polishing process present on their surface some rubber sediments that can be easily removed by any surfactant cleanser? The performed analysis by optical-digital microscopeare useful to clarify this point?
Answer: Before each measurement using the British Pendulum Tester, each specimen was thoroughly washed - corundum and fine polutions were removed. For example, the test surfaces are cleaned/washed in the same way in the FAP device (known as Wehner/Schulze), according to EN-12697-49: 2019, before measurement the coefficient of friction. In other methods that simulate surface polishing, test specimens are always washed before measuring the coefficient of friction.
Regarding to the statistical analysis, very significant differences in the wideness of the 95% confidence interval are present in the comparison between BPN values obtained respectively after Phase I and Phase II, for all the considered texturing methods. How this result can be interpreted? Why this do not happen if considering the BPN values before and after Phase I?
Answer: The reason for this phenomena is the heterogeneity of the microtexture immediately before the test. In the case of the BuD, BrD, LT and TT methods, the fine aggregate grains were exposed on the tested surface, that causes increasing of the BPN. However, there were places where the texture was smooth (cement mortar), which is characterized by low values of the friction coefficient. Hence the high values of the standard deviation.
EAC surfaces immediately after texturing are characterized by the cement mortar remaining after the exposed of the aggregate, which is removed from the aggregates along with the progressing process of polishing. This is illustrated in Figure 9e.
The conclusions actually consist of a small recapitulation of the study; the usefulness and contribution to knowledge of the proposed study should be more clearly presented.
Answer: Thank you for your attention. The conclusion has been partially changed.
Reviewer 2 Report
Dear authors,
this is mainly from the viewpoint of practice important and interesting paper which confirms some of the existing knowledge and experience which lead in the recent years to more focus on exposed aggregate concrete mainly for motorways. In general the provided scope of your study is clear and well understandable. Maybe for the statistics it would be good to somehow include what were the original expectations before analysing the data by Tukey´s honest significant difference.
In general I have only few small recommendations mainly related to the abstract.
line 12: change "Test" to "Tests"
line 12-13: delete "were tested otherwise you have too many verbs in the sentence.
line 15: I would delete "an" and "a"
line 20: "...on each stage of the test."
line 27: put comma between microtexture and macrotexture
line 77: IFI is not defined, please just put in brackets the whole term.
line 315: the heading of table 8 put on the next page
line 318-320: too many empty lines.
line 408: you are mentioning "a positive effect on the surface noise caused by cars", which in general is true and known since it has some relation to macrotexture etc. But how is this linked to the findings presented in your paper? Maybe it shall be a bit explained to support relevance of such statement in your conclusions.
Author Response
Thank you for your insightful and detailed review. Our responses and comments are provided below.
Dear authors,
this is mainly from the viewpoint of practice important and interesting paper which confirms some of the existing knowledge and experience which lead in the recent years to more focus on exposed aggregate concrete mainly for motorways. In general the provided scope of your study is clear and well understandable. Maybe for the statistics it would be good to somehow include what were the original expectations before analysing the data by Tukey´s honest significant difference.
In general I have only few small recommendations mainly related to the abstract.
line 12: change "Test" to "Tests"
Answer: Thank you for paying attention. Corrected.
line 12-13: delete "were tested otherwise you have too many verbs in the sentence.
Thank you for paying attention. Corrected.
line 15: I would delete "an" and "a"
Thank you for paying attention. Corrected.
line 20: "...on each stage of the test."
Thank you for paying attention. Corrected.
line 27: put comma between microtexture and macrotexture
Thank you for paying attention. Corrected.
line 77: IFI is not defined, please just put in brackets the whole term.
Thank you for paying attention to this detail. Corrected.
line 315: the heading of table 8 put on the next page
Thank you for paying attention. Corrected.
line 318-320: too many empty lines.
Thank you for paying attention. Corrected.
line 408: you are mentioning "a positive effect on the surface noise caused by cars", which in general is true and known since it has some relation to macrotexture etc. But how is this linked to the findings presented in your paper? Maybe it shall be a bit explained to support relevance of such statement in your conclusions.
Thank you for your attention. As the topic of noise was not discussed in the paper, this sentence was removed from the conclusions.

Reviewer 3 Report
The authors have presented a really complete work.The work is well structured and very well referenced
(with citations to numerous research papers). However,
the following corrections must be made: - Methodology: the standards used for the tests must be
cited (part 2.1.) - Table 1. It is more appropriate to represent the
granulometric distribution in a graph. -Table 2. Be careful. The Table is on 2 different pages. - It would be appropriate to introduce a section on terminology. - How many samples are prepared in each case? - Are the properties marked in Tables 2 and 3 average values?
- Figure 5.The caption is on a different page.
- The results are well argued and the tables and figures shown are enough.
- Thank you very much for the photographs.
They are very illustrative and greatly enrich the work.
Author Response
Thank you for your insightful and detailed reviews. Our responses and comments are provided below.
The authors have presented a really complete work.
The work is well structured and very well referenced (with citations to numerous research papers). However, the following corrections must be made:
- Methodology: the standards used for the tests must be cited (part 2.1.)
Answer: We agree with the Reviewer. The standards are listed in references.
- Table 1. It is more appropriate to represent the granulometric distribution in a graph.
Answer: We agree with the Reviewer. Corrected.
-Table 2. Be careful. The Table is on 2 different pages.
Answer: Thank you for paying attention to this detail.
- It would be appropriate to introduce a section on terminology.
Answer: Thank you for paying attention. A definition of skid resistance has been introduced (page 1, line 31). The definitions of microtexture and macrotexture have been defined earlier (page 2, lines: 67 - 74).
- How many samples are prepared in each case?
Answer: For each of the methods of texturing concrete pavements, 6 samples were made, 4 of which were intended for testing. Information on this can be found on level 4, line 127.
- Are the properties marked in Tables 2 and 3 average values?
Answer: Yes. Tables 2 and 3 show the mean values.
- Figure 5.The caption is on a different page.
Answer: Thank you for paying attention to this detail.
- The results are well argued and the tables and figures shown are enough.
- Thank you very much for the photographs. They are very illustrative and greatly enrich the work.

Round 2
Reviewer 1 Report
Except for the conclusions, in my opinion the paper has not been significantly improved. The answers to previous comments have been provided in the Authors' reply but they have not explained in the text of the paper.
For example, the reasons why the wideness of confidence interval of BPN values is very different in the comparison before/after tests should be an important information to provide to readers. The same for the fact that the specimens were thoroughly washed before performing BP Tests and for the fact the Slab Polisher is a proprietary device.
More in detail, I didn't catch the explanation regarding the wideness of the confidence interval: if the problem depends on the heterogeneity of the microtexture before the test, I would have expected higher intervals before the 1st phase (green line and points), while the results showed in the figure 11 seem to evidence that the higher values of the standard deviations happen after the 2nd phase (blue line and points).
Author Response
Answer to Reviewer
Except for the conclusions, in my opinion the paper has not been significantly improved. The answers to previous comments have been provided in the Authors' reply but they have not explained in the text of the paper.
For example, the reasons why the wideness of confidence interval of BPN values is very different in the comparison before/after tests should be an important information to provide to readers. The same for the fact that the specimens were thoroughly washed before performing BP Tests and for the fact the Slab Polisher is a proprietary device.
More in detail, I didn't catch the explanation regarding the wideness of the confidence interval: if the problem depends on the heterogeneity of the microtexture before the test, I would have expected higher intervals before the 1st phase (green line and points), while the results showed in the figure 11 seem to evidence that the higher values of the standard deviations happen after the 2nd phase (blue line and points).
Answer:
Thank you for paying attention again, because we found an error - the legend (description) in Figures 11 and 12 was incorrect. The first time we did not understand your remark regarding the high values of standard deviation. The highest values for both BPN and MPD were obtained before the first phase – blue colour – it is correct. Clarifications on the high BPN standard deviation have been added to the main text (line 339):” It was found that by far the highest BPN standard deviation values were obtained before the polishing phase in each of the texturing methods (Figure 11). The reason for this phenomena is the heterogeneity of the microtexture immediately before the test. In the case of the BuD, BrD, LT and TT methods, the fine aggregate grains were exposed on the tested surface, that causes increasing of the BPN. However, there were places where the texture was smooth (cement mortar), which is characterized by low values of the friction coefficient. Hence the high values of the standard deviation.
EAC surfaces immediately after texturing are characterized by the cement mortar remaining after the exposed of the aggregate, which is removed from the aggregates along with the progressing process of polishing.”
In addition, details have been added for the Slab Polisher (lines 148 and 153: “The Slab polisher is a proprietary device built at the Bialystok University of Technology.” and “A more detailed description of the Slab Polisher and examples of the results of research can be found in the literature [9, 54].”) and the polishing process (line 162: “Before each measurement using the British Pendulum Tester, each specimen was thoroughly washed - corundum and fine polutions were removed. This procedure is also performed in other devices of this type e.g. in the FAP device (known as Wehner/Schulze).”).